# The Effect of a Smartphone App with an Accelerometer on the Physical Activity Behavior of Hospitalized Patients: A Randomized Controlled Trial

**DOI:** 10.3390/s23218704

**Published:** 2023-10-25

**Authors:** Hanneke C. van Dijk-Huisman, Rachel Senden, Maud H. H. Smeets, Rik G. J. Marcellis, Fabienne J. H. Magdelijns, Antoine F. Lenssen

**Affiliations:** 1Department of Physical Therapy, Maastricht University Medical Center, 6229 HX Maastricht, The Netherlands; hanneke.huisman@mumc.nl (H.C.v.D.-H.); rachel.senden@mumc.nl (R.S.); mhh.smeets@mumc.nl (M.H.H.S.); rik.marcellis@mumc.nl (R.G.J.M.); 2Department of Internal Medicine, Division of General Medicine and Clinical Geriatric Medicine, Maastricht University Medical Centre, 6229 HX Maastricht, The Netherlands; fabienne.magdelijns@mumc.nl; 3CAPHRI School for Public Health and Primary Care, Maastricht University, 6200 MD Maastricht, The Netherlands

**Keywords:** activity monitoring, physical activity, wearable sensors, hospitalization, physiotherapy, functional recovery, mHealth

## Abstract

Inactive behavior is common in hospitalized patients. This study investigated the effectiveness of using a smartphone app with an accelerometer (Hospital Fit) in addition to usual care physiotherapy on increasing patients’ physical activity (PA) behavior. A randomized controlled trial was performed at Maastricht University Medical Centre. Patients receiving physiotherapy while hospitalized at the department of Pulmonology or Internal Medicine were randomized to usual care physiotherapy or using Hospital Fit additionally. Daily time spent walking, standing, and upright (standing/walking) (min) and daily number of postural transitions were measured with an accelerometer between the first and last treatment. Multiple linear regression analysis was performed to determine the association between PA behavior and Hospital Fit use, corrected for functional independence (mILAS). Seventy-eight patients were included with a median (IQR) age of 63 (56–68) years. Although no significant effects were found, a trend was seen in favor of Hospital Fit. Effects increased with length of use. Corrected for functional independence, Hospital Fit use resulted in an average increase of 27.4 min (95% CI: −2.4–57.3) standing/walking on day five and 29.2 min (95% CI: −6.4–64.7) on day six compared to usual care. Hospital Fit appears valuable in increasing PA in functionally independent patients.

## 1. Introduction

Physical inactivity is common during hospital stay with patients spending between 87% and 100% of their time lying in bed or seated in a chair [1,2,3,4,5]. Prolonged periods of uninterrupted sedentary behavior are common and bouts of walking are often brief [6]. This inactive behavior contributes to negative health outcomes such as functional decline [7,8], a longer length of hospital stay (LOS) [9], an increased risk of institutionalization [6,10], and mortality [7,8,11,12]. Previous studies have demonstrated that these negative health outcomes can be mitigated by enhancing patients’ physical activity (PA) behavior [13,14,15,16,17]. Therefore, it is important that effective PA-promoting interventions are implemented in the clinical care pathway of hospitalized patients [18]. 

Physiotherapy during hospital stay is primarily aimed at enhancing patients’ PA behavior and improving functional recovery of activities of daily living [2,19,20]. In order to advise patients effectively, physiotherapists need objective insight into patients’ PA behavior. Continuous PA monitoring with real-time feedback is therefore recommended within the clinical care pathway of patients [2]. Wearable activity trackers could enable this [21,22], and they also have the ability to support personalized care and empower patients in their recovery process. A recent systematic review and meta-analysis by Szeto et al. showed that interventions using wearable activity trackers in hospitalized patients are associated with higher PA levels, less sedentary behavior, and better physical functioning compared to usual care. They found a significant association between interventions using activity trackers and higher active time compared to controls (mean difference (MD) 9.75 min per day, 95% confidence interval (CI) 0.65–18.84 min per day, *I*^2^ = 87%, *p* = 0.04) [21]. 

In several studies, accelerometers are linked to a smartphone in order to gain insight into patients’ PA behavior. Van Grootel et al. investigated the preliminary effectiveness of a goal-directed PA-promoting intervention using an activity tracker in a pre–post study in patients admitted to the Department of Pulmonology or Department of Nephrology/Gastro-enterology. Data regarding patients’ PA behavior were made visible to patients and healthcare professionals via a smartphone app, a public screen on the ward, and via the electronic patient record. Before implementation, patients spent a mean total of 38 (standard deviation (SD) 21) minutes active per day. Providing insight into patients’ PA behavior and setting daily movement goals resulted in a 32% increase to 50 (SD 31) minutes active per day (*p* = 0.03) [23]. 

Furthermore, a recent non-randomized quasi-experimental study investigated the potential of using a wearable activity tracker during the physiotherapy treatment of hospitalized patients following orthopedic surgery to increase the amount of time spent standing and walking per day and the extent of functional recovery [2]. The activity tracker, Hospital Fit, is composed of an accelerometer that is connected to a smartphone app. The app contains multiple functionalities and is designed to be used in hospitalized patients. It enables objective activity monitoring, provides insight into patients’ recovery processes, and offers a tailored exercise program supported by videos. Using Hospital Fit during the postoperative physiotherapy treatment of patients undergoing orthopedic surgery resulted in an average increase of 28.4 min (95% CI: 5.6–51.3) standing and walking on postoperative day one (POD1). Patients who used Hospital Fit spent 102.99 (95% CI: 82.77–123.21) minutes standing and walking compared to 70.89 (95% CI: 58.93–82.86) minutes in patients receiving physiotherapy without Hospital Fit. Although clinical guidelines stipulating the amount of time patients should be standing and walking during hospitalization do not exist yet [3,11,24,25,26], the increase in PA behavior can be regarded as a clinically relevant contribution to mitigate negative effects of inactivity. Additionally, the odds of regaining functional recovery in activities of daily living on POD1 were 3.1 times higher (95% CI: 1.1–8.3) in patients using Hospital Fit. Despite the improvement in patients’ PA behavior and recovery process, the quasi-experimental study design might have led to a slight overestimation of the results in favor of patients who used Hospital Fit. To establish a causal relationship between Hospital Fit use and improvements in patients’ PA behavior while minimizing bias, balancing confounding factors between groups, and enhancing the internal validity, performing a randomized controlled trial (RCT) is recommended [27]. Moreover, as the median (range) LOS of patients undergoing orthopedic surgery was 4 (3–12) days, this left relatively little time to use Hospital Fit. Therefore, using Hospital Fit in a population with a longer LOS was recommended as well. 

Furthermore, patients and physiotherapists proposed to improve Hospital Fit through adding a goal setting feature, an educational feature, and sending automatically generated reminder messages. Moreover, they suggested to improve the clinical applicability of the accelerometer through being able to differentiate standing from walking as well as being able to count the number of postural transitions per day from a sedentary (lying/sitting) to an active (standing/walking) position. As a result, the accelerometer algorithm was optimized and validated in comparison to video analysis in 50 hospitalized patients under free-living conditions [19]. The improved algorithm as well as the other suggestions were then incorporated into Hospital Fit. The primary aim of this study was to investigate the effectiveness of using Hospital Fit during usual care physiotherapy in hospitalized patients on increasing patients’ PA behavior compared to usual care physiotherapy without Hospital Fit. We hypothesize that using Hospital Fit during the physiotherapy treatment of patients hospitalized at the department of Internal Medicine and the department of Pulmonology will result in an increase in the amount of PA performed compared to patients who receive usual care physiotherapy.

## 2. Materials and Methods

### 2.1. Trial Design

This is a single-center, assessor-blinded randomized controlled (1:1) superiority trial (RCT) with a parallel-group design. The study was conducted at the department of Pulmonology and the department of Internal Medicine at Maastricht University Medical Centre (MUMC+) in Maastricht, The Netherlands, between March 2021 and January 2022. The study was performed in compliance with the Declaration of Helsinki and was approved by the Medical Ethics Committee of the University Hospital Maastricht and Maastricht University (METC20–083). The study is registered at http://www.clinicaltrials.gov (accessed on 26 February 2021) (identifier NCT04797130). The CONSORT Statement was used as the reporting guideline [28].

### 2.2. Participants

Patients receiving physiotherapy while hospitalized at the department of Pulmonology or Internal Medicine were recruited by their physiotherapist and were asked for consent to be contacted by a researcher. The researcher provided patients with verbal and written information about the study. After a 24 h interval, the researcher reached out to the patient once more and secured written informed consent before commencing the study. Data processing confidentiality and participant anonymity were assured. 

Patients were eligible if they met the following inclusion criteria: aged between 18 and 75 years, receiving physiotherapy while hospitalized at the department of Pulmonology or Internal Medicine, able to walk independently two weeks before admission as reported on the Functional Ambulation Categories scale (FAC > 3) [29,30], sufficient understanding of the Dutch language, and having access to a smartphone. Exclusion criteria were the following: presence of contraindications to walking or wearing an accelerometer on the upper leg, mentally incapacitated subjects, cognitive impairment (dementia/delirium) as reported by the attending physician, admission to the intensive care unit, a life expectancy of less than three months, and previous participation in the current study.

### 2.3. Randomization, Blinding, and Treatment Allocation

After written consent was obtained, patients were randomly allocated to a control or intervention group by a blinded researcher. Randomization was performed concealed using a computerized random number generator provided by Castor v2023.3.2.2 EDC online software [31]. Stratified block randomization was used with four computer-generated blocks, an allocation ratio of 1:1, and stratification per department (Pulmonology or Internal Medicine). Blinding patients and physiotherapists to the treatment allocation was not feasible due to the nature of the intervention. A researcher who was blinded for allocation performed the data analysis.

### 2.4. Study Procedures and Intervention

All patients had been referred to usual care physiotherapy by their physician. Physiotherapy sessions focused on enhancing PA and promoting functional independence in activities of daily living crucial for independent living at home, such as walking and climbing stairs. The specific content of the physiotherapy sessions varied based on the diagnosis and individual requirements of the patients. 

Patients allocated to the control group received usual care physiotherapy. Their PA behavior was monitored with an accelerometer, but they did not receive any feedback. Patients allocated to the intervention group used Hospital Fit additionally.

#### 2.4.1. Device Description

PA behavior was monitored using the MOX activity monitor (MOX; Maastricht Instruments B.V., The Netherlands (Figure 1)). This device featured a tri-axial accelerometer sensor (ADXL362; Analog Devices, Norwood, MA, USA) enclosed within a compact, waterproof casing measuring 35 × 35 × 10 mm and weighing 11 g. Raw acceleration data (±8 g) were recorded along three orthogonal sensor axes (X, Y, and Z) at a sampling rate of 25 Hz. The sensor was factory-calibrated against gravity for all three axes. The algorithm has been validated to distinguish between dynamic and sedentary activities and to detect postural transitions in hospitalized patients under free-living conditions [19]. The classified results were transmitted from the accelerometer to the Hospital Fit app every minute via a Bluetooth protocol. 

#### 2.4.2. Intervention

Hospital Fit is composed of a smartphone app that is linked to a MOX activity monitor through Bluetooth connectivity. The app contains multiple functionalities (Figure 2). First, a real-time summary of PA offers patients and their physiotherapists immediate insight into the number of minutes per day spent walking, standing, or sedentary (lying/sitting); the number of postural transitions per day from a sedentary to an active (standing/walking) position; the number of bouts per day spent walking for ≥5 min; and the number of bouts per day spent sedentary for ≥30 min. This enables physiotherapists to optimally coach patients in enhancing time spent walking and reducing long periods of uninterrupted sedentary behavior. To support this, Hospital Fit contains the option of setting a walking goal regarding the desired number of minutes to be spent walking. 

Second, a recovery overview provides patients insight into their recovery progress. The recovery progress can be evaluated in the app by the physiotherapist during every session using the modified Iowa Level of Assistance Scale (mILAS) [32]. The mILAS assesses the amount of assistance and type of walking aid needed in order to perform basic activities of daily living (i.e., moving from a supine position to a seated position and back, moving from a seated to a standing position, walking, and climbing stairs). If necessary, the mILAS score can be adapted multiple times per day. The mILAS score was transformed into a percentage score in the app, with 100% reflecting functional independence. Percentage scores are shown per activity, providing patients with insight into which activities need improvement to reach functional independence [2]. 

Third, Hospital Fit enables physiotherapists to create a personalized exercise program. The app contains a database of 25 videos aimed at strengthening the upper and lower limbs, enhancing physical fitness and functional recovery. The physiotherapist can select videos supporting patients’ treatment goals and self-management. The exercise program can be adapted as often as preferred and the physiotherapist can add notes regarding the number of repetitions or sets. 

Lastly, Hospital Fit contains an educational video informing patients of the importance of remaining as active as possible during hospitalization.

#### 2.4.3. Hospital Fit Procedures

During the first treatment, the physiotherapist assisted with installing the Hospital Fit app on the patient’s smartphone and provided verbal and written information on the main functionalities of Hospital Fit. In accordance with the patient, the physiotherapist set a personalized goal in the app concerning the target number of minutes of daily walking. As there are currently no guidelines specifying the recommended daily duration of walking for hospitalized patients [3,11,24,25,26], the goal was individually determined and was influenced by factors including the patient’s diagnosis, symptoms, and specific healthcare needs. Additionally, the physiotherapist rated the amount of assistance and type of walking aid needed during activities of daily living based on the mILAS and created a personalized exercise program based on patients’ individual requirements. During every consecutive physiotherapy session, the walking goal, mILAS score, and exercise program were evaluated with the patient and adapted if necessary. Patients were instructed by their physiotherapist to use Hospital Fit as often as they deemed necessary but at least once daily. No strict protocol was provided regarding the number of times patients should open the app. Four times a day, an automatically generated notification message was sent, serving as a reminder to open the app. Furthermore, data collected in the PA overview and recovery overview were reported in the patient’s electronic medical record once per day, enabling other healthcare professionals (e.g., nurses and physicians) to use the information to coach patients in their PA behavior and recovery. Nurses and physicians were instructed to evaluate patients’ PA behavior and recovery during daily rounds.

### 2.5. Outcomes

#### 2.5.1. Physical Activity

The primary outcome measure was time spent walking per day (minutes). Secondary outcome measures were time spent standing per day (minutes), time spent upright (standing/walking) per day (minutes), and number of postural transitions from a sedentary position (lying/sitting) to an active position (standing/walking) per day.

During the first physiotherapy session after inclusion, the accelerometer was fixated to the right anterior thigh (ten centimeters above the patella) using a hypoallergenic patch (Figure 3). Each day, the skin around the patch was inspected by a nurse for signs of irritation. PA behavior was monitored 24 h per day. The monitoring of PA ended after a week or on the day of discharge, depending on which occurred first. Participation in the study ended after removing the accelerometer and uploading the data to a computer. A complete measurement day was defined as a 24 h period beginning or ending at midnight. Days with at least 20 h of recorded wear time were deemed valid measurement days and were included in the data analysis. Subsequently, the primary and secondary outcome measures were calculated per patient.

#### 2.5.2. Functional Independence

Functional independence was evaluated by the physiotherapist during every treatment session using the mILAS and was documented in the electronic health record. In the intervention group, the mILAS was also documented in the app. The mILAS evaluates a patient’s ability to perform various activities of daily living, including transferring from a supine position to a seated position and vice versa, sitting to standing, walking, and climbing stairs. It rates the level of assistance and type of walking aid required for each task, with scores ranging from 0 to 6 points per item. Total scores ranged from 0 to 30, with zero indicating independence on all items. Stair climbing was assessed only if patients had to perform this at home; otherwise it was scored as zero [32]. The total mILAS score was dichotomized into two groups: functional dependence (0: mILAS score > 0) versus functional independence (1: mILAS score = 0). The mILAS has demonstrated high reliability, validity, and responsiveness when used to asses functional independence in hospitalized patients [32]. 

#### 2.5.3. Medical and Demographic Data

The following medical and demographic outcome measures were extracted from the electronic health record by the researcher: age (years), sex (male/female), number of physiotherapy treatment sessions received during study participation (n), LOS (days), and discharge location (home, geriatric rehabilitation center, nursing home, other).

### 2.6. Sample Size 

Sample size was calculated via the online sample size calculator ClinCalc.com [33]. Based on a significance level of 0.05, a power of 0.80, an effect size of 0.5, and two determinants (Hospital Fit use and functional independence (dichotomized mILAS score)), a sample size of *n* = 66 was required. The effect size was based on previous studies with a similar study population and intervention [2,34]. Accounting for a 15% drop-out rate, we aimed to enroll *n* = 78 patients in this study. 

### 2.7. Data Analysis

The data underwent a thorough examination to ensure completeness and identify any inconsistencies. Stochastic regression imputation with fully conditional specification was used to impute missing values in case ≥ 5% of the data was missing. Descriptive statistics are presented as means with 95% confidence intervals (CIs) for normally distributed continuous variables or as medians with interquartile ranges (IQRs) for not normally distributed data. Categorical data were summarized by numbers (*n*) and percentages (%). Univariate regression analysis was conducted to assess the association between time spent walking per day and Hospital Fit use (group). Subsequently, a multiple linear regression analysis was carried out to assess the association between time spent walking per day and Hospital Fit use, while adjusting for functional independence (dichotomized mILAS score) as a potential confounding variable. Functional independence was included in the regression model when it resulted in a change of ≥10% in the regression coefficient of the main determinant (Hospital Fit use) [35]. For all secondary outcome measures, the same procedure was performed and the same confounding variable (functional independence) was analyzed. Data were analyzed according to the intention-to-treat principle. Missing values were not substituted and drop-outs were not replaced. For all statistical analyses, the level of significance was set at *p* < 0.05. Data were analyzed using SPSS (version 28.0.0.0; IBM Corporation, Armonk, NY, USA).

## 3. Results

A total of 757 patients were admitted to the department of Pulmonology or Internal Medicine and were screened for eligibility, resulting in 289 eligible patients. Among them, 78 patients were included in this study, with 39 patients randomly allocated to the intervention group and 39 to the control group (Figure 4).

Participants had a median (IQR) age of 63 (56–68) years and 44 (56%) of them were male. Characteristics of the study participants are presented in Table 1. 

Of the included patients, two (3%) dropped out due to a decline in health. Data of 76 (97%) patients were used in the analysis, of which 38 patients (50%) were in the intervention group and 38 (50%) were in the control group (Figure 4). PA data were missing for 19 out of 284 valid measurement days (7%): 7 days were missing in the control group (3%) and 12 days in the intervention group (4%). Missing PA data were spread over eight patients (11%): two in the control group (3%) and six in the intervention group (8%). Reasons for this missing PA data were unjustified removal of the accelerometer (*n* = 4), removing the accelerometer due to an MRI (*n* = 1), accelerometer malfunctioning (*n* = 2), and accelerometer falling off (*n* = 1). mILAS scores were missing for 33 days (12%): 23 days were missing in the control group (8%) and 10 days in the intervention group (4%). Missing mILAS scores were spread over 13 patients (17%): 8 patients in the control group (10%) and 5 patients in the intervention group (7%). The reason for missing mILAS scores was lack of reporting by the physiotherapist. After imputation, data of all 76 patients were complete for analysis.

PA data were collected for a median (IQR) of 4 (2–6) valid measurement days. On day two, PA data were available for 76 patients (100%). On day three, eight patients were discharged, and PA data were available for 68 patients (90%), with 33 in the control group and 35 in the intervention group. During the following days, more patients were discharged and the available PA data per day continued to decrease until day seven (Table 2). The distribution of functional independence between groups is shown in Appendix A, Table A1.

The results of the univariate linear regression analyses showed that time spent walking ranged between 32.5 and 40.6 min in the control group and between 35.9 and 45.5 min in the intervention group. Time spent upright ranged between 58.4 and 69.9 min in the control group and between 64.8 and 77.4 min in the intervention group. Hospital Fit use did not result in a significant increase in time spent walking per day, time spent standing per day, time spent upright per day, or the number of postural transitions from a sedentary to an active position from day two to seven (Appendix B–Table A2, Table A3, Table A4 and Table A5). 

Next, multiple linear regression analyses were conducted to correct for the potential influence of functional independence (independence on dichotomized mILAS) on the association between Hospital Fit use and PA behavior. The results of the multiple linear regression analyses of the association between Hospital Fit and time spent walking per day are shown in Table 2. Hospital Fit use led to an increase in time spent walking per day compared to the control group, with 17.1 (95% CI: −0.8–34.9) minutes more on day five and 15.6 (95% CI: −4.6–35.8) minutes more on day six. Although a trend was seen, the association between Hospital Fit use and time spent walking remained non-significant on all days. 

The multiple linear regression analyses of the association between Hospital Fit use and time spent upright per day also showed an increase in time spent upright per day with Hospital Fit use (Table 3). On day five and six, patients spent 27.4 (95% CI: −2.4–57.3) and 29.2 (−6.4–64.7) more minutes upright compared to patients in the control group, respectively. Despite the positive trend, the association between Hospital Fit use and time spent upright remained non-significant on all days. Multiple linear regression analyses of all other secondary outcome measures also showed that the addition of functional independence increased the effect of Hospital Fit use without reaching a significant effect (Appendix C, Table A6 and Table A7). In all multiple regression analyses, the effect of Hospital Fit use increased the longer patients used Hospital Fit.

The sensitivity analysis demonstrated comparable findings between the imputed and the original non-imputed datasets, leading us to conclude that imputation did not lead to large differences.

## 4. Discussion

This study evaluated the effectiveness of Hospital Fit in addition to usual care physiotherapy on enhancing the PA behavior of hospitalized patients. It was hypothesized that patients who used Hospital Fit would spend more time walking per day than patients that received usual care physiotherapy without using Hospital Fit. Moreover, an increase in time spent standing per day, time spent upright per day, and an increase in the number of daily postural transitions from a sedentary to an active position were expected with Hospital Fit use. Although patients who used Hospital Fit spent more time standing and walking and interrupted their sedentary behavior more often compared to patients that received usual care physiotherapy without using Hospital Fit, the effects were not significantly different between the groups, even after correcting for the influence of functional independence.

The hypothesis that Hospital Fit use results in an increase in PA behavior of hospitalized patients was based on a previous study that demonstrated an average increase of 28.4 min (95% CI: 5.6–51.3) standing/walking on POD1 after the introduction of Hospital Fit to the usual care physiotherapy in patients following orthopedic surgery [2]. In the current study, a trend was seen showing an increase in all outcome measures with Hospital Fit use. The effects increased with the number of days patients used Hospital Fit. On day two and three, the effects of Hospital Fit use were very small. This may have been caused by the Hawthorne effect. Patients in the control group were wearing an accelerometer without receiving feedback on their PA behavior. However, they were aware that their activity levels were being monitored. As a result, patients in the control group may consciously or subconsciously have also increased their PA behavior in response to wearing an accelerometer. The Hawthorne effect tends to have a more pronounced impact in the short term and may fade over time, resulting in larger differences in PA behavior between the control and intervention group on following days. Moreover, patients may need some time to become familiarized with the use of Hospital Fit and to receive feedback from their physiotherapist regarding their PA behavior. It is possible that patients may need to use Hospital Fit for a longer period before a significant difference in their PA behavior occurs.

On day five and six, the largest effects were seen. When corrected for functional independence, patients who used Hospital Fit spent on average 27.4 min (95% CI: −2.4–57.3) more standing and walking on day five and 29.2 min (95% CI: −6.4–64.7) more on day six than patients who did not use Hospital Fit. Although these results are not significant, they are comparable to the results found in the previous study investigating Hospital Fit in orthopedic patients [2]. Moreover, a recent systematic review and meta-analysis by Szeto et al. found a mean difference of 9.75 active minutes per day in favor of interventions that used wearable activity trackers to increase the PA behavior of hospitalized patients compared to usual care. They described that this increase in PA behavior already makes a considerable impact on patient recovery during hospitalization [21]. Furthermore, Grootel et al. showed a significant effect in their goal-directed movement intervention with an increase from 38 (SD 21) to 50 (SD 31) minutes active per day (*p* = 0.03). They indicate that the 12 min increase in active time plays a clinically relevant role towards reducing the risk of functional decline and postoperative complications [26]. In the current study, similar amounts of time spent upright are seen from day four of Hospital Fit use onwards. Although the results are not statistically significant, the observed trends can be considered clinically relevant towards the prevention of negative health outcomes associated with inactivity.

The PA behavior of patients included in the current study was comparable to other studies. Pitta et al. showed that patients admitted for an exacerbation of Chronic Obstructive Pulmonary Disease (COPD) spent a median time of 50.4 (IQR 21.6–133.9) minutes upright on day two and 64.8 (IQR 50.4–151.2) minutes on day seven of hospitalization [36]. Brown et al. found that patients admitted to a medical ward spent on average 54.7 (SD 50.4) minutes per day upright during the first week of hospitalization [37]. Dall et al. studied the effect of providing patients with feedback on their PA behavior on time spent walking, standing, sitting, and lying in patients admitted to a Pulmonology ward [38]. On average, patients receiving no feedback spent 64 (95% CI −3–131) minutes upright per day. Although patients receiving feedback spent 18 (95% CI −42–78) minutes more upright per day, no significant group differences were found in time spent upright and other PA outcome measures. However, when corrected for the influence of functional independence, they found that patients with independent walking ability spent 51 min (95% CI 0–102) more upright when they received visual feedback in contrast with patients that received no feedback. Dall et al. suggested that providing feedback could be beneficial in promoting PA behavior when provided to patients with independent walking abilities [38], which is supported by the results of the current study.

### 4.1. Strengths and Limitations

To our knowledge, the current study is the first to explore the effectiveness of a multimodal mHealth tool on the PA behavior of patients receiving physiotherapy during hospitalization at the departments of Pulmonology or Internal Medicine. This study has a number of strengths, amongst which are the use of a randomized controlled study design and correcting for missing data. Missing data related to PA monitoring could affect the accuracy of the reported outcomes, and missing data related to functional dependence (mILAS scores) could affect the adjustment for this potential confounding variable and lead to biased results. To correct for the impact of missing data, stochastic regression imputation was performed. While imputation introduces some uncertainty, it enhances the completeness of the dataset, potentially reducing biases and leading to more reliable results. We have accounted for the uncertainty introduced by imputation by providing confidence intervals and conducting sensitivity analyses. Moreover, we also acknowledge some limitations. First, we did not monitor how often patients used Hospital Fit or which functionalities patients used. Therefore, we cannot establish a relationship between the frequency of using Hospital Fit and its different functionalities on patients’ PA behavior. The effect of Hospital Fit may have been influenced by patient compliance and engagement. If patients did not consistently use Hospital Fit or did not follow the personalized exercise programs, this could have limited the effectiveness of the intervention. Second, a heterogeneous patient population was included in this study, consisting of patients admitted for acute as well as elective reasons with a variety of diagnoses, illness severities, and symptoms. This heterogeneity may have introduced variations in the outcomes and responses to Hospital Fit. Differences in diagnoses may have led to variations in baseline health, varying mobility levels and different treatment plans, potentially influencing the effect of Hospital Fit. Moreover, illness severity and symptoms (e.g., pain, fatigue, dyspnea) may have influenced patients’ ability to engage in PA. Additionally, some patients were dependent on healthcare professionals to assist them during walking and their PA behavior may have been influenced by a shortage in staffing. In the current study, we did not correct for these potentially confounding factors. Third, selection bias may have been introduced as patients with a lower motivation to be physically active or a higher illness severity may not have been willing to participate in this study. Fourth, the current study was powered to detect an effect with 66 patients. Although sufficient patients were included on day two and three, a number of patients were discharged every day. While clinically relevant effects can be seen from day four on, we lack the power to reach significance. We expect that if we had had sufficient power on days four to seven, we might have been able to detect a significant effect of Hospital Fit use. Lastly, patients received a median (IQR) number of 2 (2–3) physiotherapy sessions. For some patients, this may not have been sufficient to create an effect of using Hospital Fit.

### 4.2. Clinical Implications and Recommendations for Future Research

The present study has a number of implications for clinical practice and future research. First, it provides valuable insight that the effect of Hospital Fit increases with the number of days patients use Hospital Fit and suggests that a clinically relevant contribution to the prevention of the negative effects of inactivity can be seen when Hospital Fit is used for at least four days. The increase in PA behavior has a positive influence on the prevention of functional decline and complications, a reduction in LOS, an improved quality of life, and reduced risk of institutionalization. Moreover, the positive trend in the increase in PA with Hospital Fit also supports personalized care and empowers patients in their recovery process. Second, the results demonstrate a strong association between functional independence and PA, with independent patients spending significantly more time standing and walking per day and performing more transitions per day than patients that are dependent on others. Patients that were dependent on others and did not use Hospital Fit spent on average 4.3 min walking on day five and 5.2 min on day six, compared to 41.4 and 39.2 min in independent patients that did not use Hospital Fit. In independent patients using Hospital Fit, time spent walking on day five and six increased to 58.4 and 54.8 min, respectively. In order to prevent the negative effects of inactivity, patients that are dependent on others should therefore first receive intensive treatment to regain functional independence. Once independence is regained, patients may benefit from PA-promoting interventions such as Hospital Fit. 

Lastly, the data created by Hospital Fit have tremendous potential. Monitoring PA as usual care enables physiotherapists to make more efficient choices regarding which patients need treatment, but it also enables creating population norms for PA. But before implementing Hospital Fit in standard care, we would first propose to study the effect of Hospital Fit in patients that are functionally independent and are able to use Hospital Fit for at least four days, in a study that is adequately powered on all consecutive measurement days. Additionally, 45% of the patients that were screened for eligibility in the present study were excluded due to their age exceeding 75 years. As we see an increase in smartphone use amongst elderly patients, we propose to remove this exclusion criterion in future studies. Moreover, the primary focus of the current study was to investigate the effectiveness of using Hospital Fit on increasing patients’ PA behavior. In future studies, we propose to perform a process evaluation as well, to investigate how often Hospital Fit and the different functionalities were used and to allow a correct interpretation of the effectiveness. 

## 5. Conclusions

This study aimed to investigate the effectiveness of using Hospital Fit during usual care physiotherapy in hospitalized patients in increasing patients’ PA behavior compared to usual care physiotherapy without using Hospital Fit. Although no significant effects were found, a trend was seen in favor of patients who used Hospital Fit. The effect increased with the number of days Hospital Fit was used with an average increase of 27.4 min (95% CI: −2.4–57.3) standing/walking on day five and 29.2 min (95% CI: −6.4–64.7) on day six compared to usual care when corrected for functional independence. Additionally, a strong association was seen between functional independence and patients’ PA behavior. Considering its clinically relevant contribution to promoting PA behavior in hospitalized patients, preventing negative health outcomes associated with inactive behavior, supporting personalized care, and empowering patients, we believe that Hospital Fit appears to be a valuable tool in addition to usual care physiotherapy for functionally independent patients. For future research, it is recommended to study the effect of Hospital Fit solely in patients that are functionally independent and are able to use Hospital Fit for at least four days, in a study that is adequately powered on all consecutive measurement days.

## Figures and Tables

**Figure 1 sensors-23-08704-f001:**
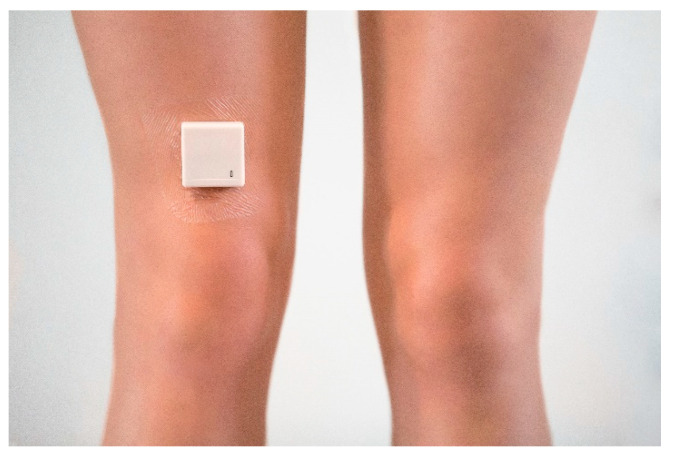
The MOX activity monitor.

**Figure 2 sensors-23-08704-f002:**
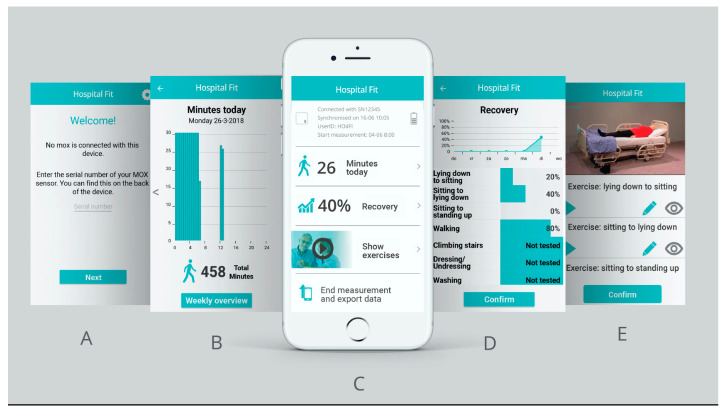
Overview of the main functionalities of Hospital Fit. (**A**) Login screen; (**B**) physical activity summary; (**C**) main menu; (**D**) recovery overview; (**E**) exercise program.

**Figure 3 sensors-23-08704-f003:**
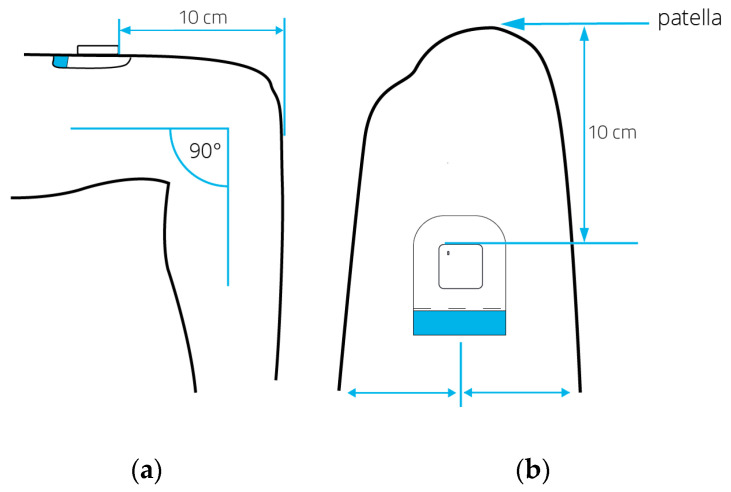
Positioning of the MOX activity monitor from a lateral view (**a**) and frontal view (**b**) while the patient is seated. The arrows highlight the placement of the hypoallergenic patch and activity monitor on the upper thigh, situated 10 cm proximal of the patella.

**Figure 4 sensors-23-08704-f004:**
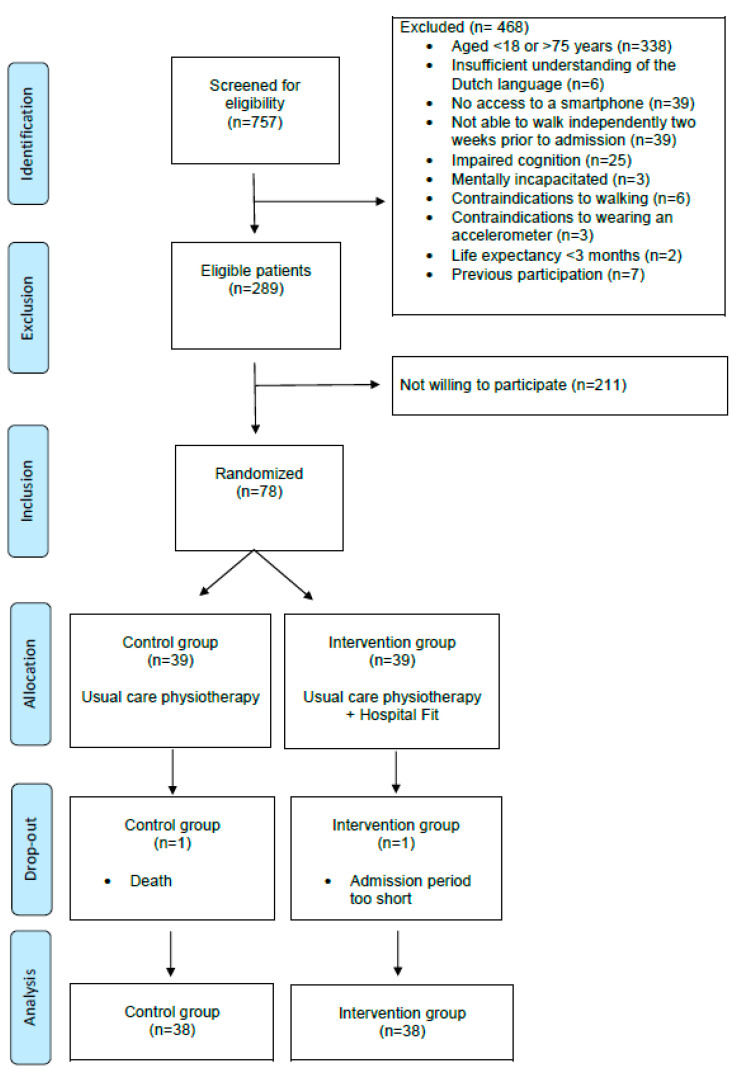
CONSORT flow chart.

**Table 1 sensors-23-08704-t001:** Characteristics of study participants.

	Control Group (*n* = 39)	Intervention Group (*n* = 39)
Age, years (median, IQR)	62 (55–67)	64 (56–69)
Sex (*n*, %)		
Female Male	18 (46) 21 (54)	16 (41) 23 (59)
Department (*n*, %)		
Pulmonology Internal Medicine	26 (67) 13 (33)	26 (67) 13 (33)
Walking aid use before admission (*n*, %)		
No walking aid Walker Cane Other	28 (71) 5 (13) 5 (13) 1 (3)	29 (74) 9 (23) 1 (3) 0 (0)
Number of PT sessions received during participation	2 (1–3)	2 (2–3)
LOS, days (median, IQR)	10 (6–18)	9 (6–25)
Discharge location (*n*, %)		
Home Geriatric rehabilitation center Nursing home Other	31 (79) 4 (10) 1 (3) 3 (8)	31 (79) 4 (10) 1 (3) 3 (8)

Characteristics of patients included in the study, categorized by group. IQR = Interquartile Range, PT = Physiotherapy, LOS = length of hospital stay.

**Table 2 sensors-23-08704-t002:** Multiple linear regression analyses—the association between time spent walking per day (minutes) and Hospital Fit use.

Day	*n* (%)		B	Std. Error	*p*-Value	95% Confidence Interval for B	
						Lower Bound	Upper Bound
2	76 (100)	Constant	33.02	5.21	<0.001	22.65	43.40
		Hospital Fit use	2.88	7.37	0.697	−11.78	17.55
3	68 (89.5)	Constant	16.84	7.20	0.022	2.45	31.22
		Hospital Fit use	5.99	7.06	0.399	−8.11	20.09
		Functional independence (independent)	28.94	7.38	<0.001	14.19	43.68
4	54 (71.1)	Constant	12.02	8.11	0.145	−4.27	28.31
		Hospital Fit use	8.38	7.57	0.273	−6.81	23.57
		Functional independence	38.04	8.28	<0.001	21.42	54.67
5	44 (57.9)	Constant	4.26	9.74	0.664	−15.40	23.93
		Hospital Fit use	17.07	8.85	0.061	−0.80	34.94
		Functional independence	37.10	9.69	<0.001	17.54	56.67
6	34 (44.7)	Constant	5.21	10.95	0.638	−17.13	27.54
		Hospital Fit use	15.62	9.89	0.124	−4.56	35.80
		Functional independence	33.94	10.57	0.003	12.38	55.51
7	27 (35.5)	Constant	14.05	12.77	0.282	−12.31	40.41
		Hospital Fit use	9.56	12.36	0.447	−15.95	35.07
		Functional independence	29.89	12.57	0.026	3.95	55.83

**Table 3 sensors-23-08704-t003:** Multiple linear regression analyses—the association between time spent upright (standing/walking) per day (minutes) and Hospital Fit use.

Day	*n* (%)		B	Std. Error	*p*-Value	95% Confidence Interval for B	
						Lower Bound	Upper Bound
2	76 (100)	Constant	60.14	8.87	<0.001	42.46	77.82
		Hospital Fit use	5.92	12.55	0.638	−19.08	30.92
3	68 (89.5)	Constant	34.68	12.92	0.009	8.88	60.48
		Hospital Fit use	8.57	12.66	0.501	−16.72	33.86
		Functional independence	47.11	13.24	<0.001	20.67	73.56
		(independent)					
4	54 (71.1)	Constant	28.77	12.99	0.031	2.70	54.83
		Hospital Fit use	11.76	12.11	0.336	−12.55	36.07
		Functional independence	54.87	13.25	<0.001	28.26	81.47
5	44 (57.9)	Constant	15.82	16.25	0.336	−17.00	48.64
		Hospital Fit use	27.43	14.77	0.071	−2.40	57.26
		Functional independence	52.41	16.17	0.002	19.75	85.06
6	34 (44.7)	Constant	14.92	19.30	0.445	−24.44	54.28
		Hospital Fit use	29.17	17.43	0.104	−6.39	64.72
		Functional independence	52.85	18.63	0.008	14.85	90.85
7	27 (35.5)	Constant	26.42	19.96	0.198	−14.78	67.61
		Hospital Fit use	18.53	19.31	0.347	−21.32	58.39
		Functional independence	46.35	19.64	0.027	5.82	86.88

## Data Availability

Additional data supporting the findings of this work are available upon reasonable request by contacting the corresponding author.

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
