# Peer review of "The Effect of a Smartphone App with an Accelerometer on the Physical Activity Behavior of Hospitalized Patients: A Randomized Controlled Trial"

_sensors, 2023, doi:10.3390/s23218704_

Round 1
Reviewer 1 Report
This study “The Effect of a Smartphone App with Accelerometer on the Physical Activity Behavior of Hospitalized Patients: a Randomized Controlled Trial” conducted by Huisman et al., examined the impact of a smartphone app, Hospital Fit, equipped with an accelerometer, on the physical activity levels of hospitalized patients in addition to standard physiotherapy. A randomized trial involving patients from Pulmonology and Internal Medicine departments revealed no significant effects, but a positive trend in favor of Hospital Fit use. Effects were more pronounced with longer app usage, particularly on days five and six. The study also highlighted a strong link between functional independence and physical activity behavior, suggesting the potential value of Hospital Fit for functionally independent patients.
The sample size in the study is adequate and enough to draw the conclusion for a knowledgeable reader. This study will be useful for the researchers working in this field and could help them advance their knowledge in designing physical activity monitoring apps. The results are comprehensive and presented in sound and reader-friendly format. Therefore, I recommend this paper for publication in this current format in the prestigious journal “Sensors”
Reviewer 2 Report
The Effect of a Smartphone App with Accelerometer on the Physical Activity Behavior of Hospitalized Patients: a Randomized Controlled Trial
Comments:
Introduction
-
52-55: While the results of the pilot study are mentioned, specific data regarding the increases in standing and walking time are not provided, and key findings are not discussed in detail. It would be appropriate to mention specific limitations of the pilot study, such as sample size or potential bias. This would provide a more comprehensive understanding of the validity of the results.
-
It is necessary to include references to multiple related studies and analyze their results to provide a more comprehensive context for the findings.
-
63-65: It should be explained why a randomized controlled trial is essential for validating the findings and providing robust evidence.
-
The improvement suggestions for Hospital Fit and the accelerometer algorithm are mentioned, but detailed information on how they were optimized and validated is not provided.
Materials and methods
-
106-114: Provide a brief but clear explanation of the importance of randomization in clinical trials and how it helps ensure unbiased allocation.
-
109: When mentioning the use of Castor EDC online software for randomization, consider adding a brief description or reference for readers who may want to learn more about the tool.
-
143: Provide a concise yet informative breakdown of the multiple functionalities of the app. For example, explain why tracking minutes spent in different activities and postural transitions is relevant to patient care.
-
190: Provide reasons for selecting 20 hours as the threshold for valid measurement days.
-
221-227: Provide a brief justification for the chosen parameters used in the sample size calculation, including the significance level, power, and effect size, to enhance the understanding of their relevance in the context of this study.
Results
-
250: Justify the chosen sample size of 78.
-
265-277: Provide a sentence discussing the potential impact of missing data on the study's results.
-
285-292: Discuss the potential reasons for the lack of significance in the results. Were there any unexpected findings, and how do these results compare to prior expectations or similar studies in the literature?
-
293-301: While the paragraph mentions a trend, it would be beneficial to briefly discuss the significance of this trend and why it may be important, even if it did not reach statistical significance.
Discussion
-
Emphasize the observed trends in the results, even if they were not statistically significant. Discuss the potential reasons for these trends and their clinical significance.
-
The study mentions the Hawthorne effect as a possible reason for small effects on days two and three. Provide more detail on what the Hawthorne effect is and how it might have influenced the results.
-
Discuss the limitation of having a heterogeneous patient population with various diagnoses and illness severities. Recognize that this diversity might introduce variability in PA behavior and acknowledge that the study did not correct for these potential confounding factors.
Conclusions
-
The conclusion could emphasize the clinical significance of the observed trends, even if they did not reach statistical significance. While no significant effects were found, highlighting the potential positive impact on patient care, especially in the context of preventing negative health outcomes associated with inactivity, can strengthen the conclusion.
-
Consider providing a broader reflection on the overall impact and implications of this research in the context of patient care and hospital protocols. What does this study mean for the future of mHealth tools like Hospital Fit in clinical settings?
Minor editing of English language required
Reviewer 3 Report
Reviewer's Report:
Title:
The title of the manuscript, "The Effect of a Smartphone App with Accelerometer on the Physical Activity Behavior of Hospitalized Patients: a Randomized Controlled Trial," is clear and informative, providing a succinct description of the study's focus.
Abstract:
The abstract provides a concise overview of the study's objectives, methods, and some key findings. However, there are areas that need improvement:
1. Lack of Specific Results: The abstract mentions a "trend in favor of Hospital Fit use" but does not provide specific quantitative data or effect sizes related to physical activity outcomes. Providing such data would make the abstract more informative.
2. Clarity of Impact: The abstract mentions that effects were "largest on days five and six" but does not specify the magnitude of these effects. Quantifying the increase in physical activity on these days would enhance the abstract's clarity.
3. Limited Mention of Limitations: The abstract does not touch upon any limitations of the study. Including a brief mention of potential limitations would provide a more balanced perspective to readers.
Introduction:
The introduction section offers valuable context and background information regarding the problem of physical inactivity in hospitalized patients. However, there are a few points:
1. Hypothesis Statement: The introduction does not explicitly state the study's hypothesis or research question. Clearly outlining the research question or hypothesis would help readers understand the study's purpose.
2. Supporting Statistics: While the introduction mentions the health outcomes associated with inactivity, it lacks specific statistics or references to studies that support these claims. Providing concrete data or citations would strengthen the introduction's argument.
3. Lack of Mention of Previous Research: Although the introduction refers to previous studies, it does not cite or discuss specific studies or findings related to the use of smartphone apps with accelerometers in hospitalized patients. Incorporating references to relevant prior research would add depth to the introduction.
Methods:
The methods section outlines the study design, participant recruitment, interventions, and outcome measures. However, there are several areas that need improvement:
1. Lack of Detail: The methods section lacks sufficient detail on key aspects of the study, such as the randomization process, the criteria for patient inclusion and exclusion, and the specific procedures for monitoring physical activity and functional independence. More detailed information is needed for transparency and reproducibility.
2. Terminology Explanation: The term "mILAS" is introduced without prior explanation. It should be defined or briefly described when first mentioned to ensure readers understand its relevance to the study.
3. Clarity of Data Analysis: The methods section briefly mentions multiple linear regression analysis but does not provide details about the variables included in the analysis, the statistical significance level, or how the analysis was corrected for potential confounding variables. Adding these details would enhance the clarity of the statistical approach used in the study.
4. Intervention Description: While the methods describe the Hospital Fit intervention, it could benefit from more information on how patients were instructed to use the app, how often they were encouraged to interact with it, and whether there were any specific goals or feedback mechanisms in place. Providing these details would help readers understand the intervention better.
Results Section
1. Sample Size and Power: The manuscript mentions that the study was powered to detect an effect with 66 patients, but the sample size in the intervention and control groups is reported as 39 each. However, it is unclear why the sample size fell short of the initially planned number, and this discrepancy should be addressed. The study's underpowered nature may have influenced the ability to detect significant effects, which should be discussed.
2. Missing Data: The manuscript mentions missing data for PA (physical activity) measurements and mILAS scores. It is crucial to provide a clear explanation of how missing data were handled in the analysis, especially after imputation. The rationale for the imputation method used and its potential impact on the results should be discussed.
3. Heterogeneous Patient Population: The manuscript acknowledges that the patient population was heterogeneous in terms of diagnoses, illness severity, and symptoms. Given this heterogeneity, it is essential to discuss how these variations might have affected the outcomes and whether specific subgroups responded differently to Hospital Fit.
Discussion Section:
4. Clinical Relevance: While the manuscript discusses trends in PA behavior improvement with Hospital Fit, it is essential to emphasize the clinical relevance of these trends. Did the observed trends translate into meaningful improvements in patients' physical activity levels and health outcomes? The discussion should include a more explicit exploration of the clinical implications of the findings.
5. Comparison to Previous Studies: The manuscript cites previous studies on similar topics, but it would be beneficial to provide a more comprehensive discussion of how the current findings align or differ from existing literature. This would help readers better understand the significance of the results.
6. Process Evaluation: The manuscript suggests conducting a process evaluation in future studies to investigate how often Hospital Fit and its functionalities were used. While this is a reasonable suggestion, it would be valuable to discuss why such an evaluation was not included in the current study and how it might have contributed to the interpretation of the findings.
7. Recommendations for Future Research: The manuscript provides recommendations for future research, which is good practice. However, these recommendations could be more specific and tailored to addressing the limitations and unanswered questions raised by the current study.
nil
Round 2
Reviewer 2 Report
In the reviewed version of "The Effect of a Smartphone App with Accelerometer on the Physical Activity Behavior of Hospitalized Patients: a Randomized Controlled Trial," the authors have made commendable efforts to address the issues present in the draft version. They have not only resolved these concerns but have also introduced new and supportive references throughout the paper. In this regard, I find the article to be acceptable in its current form.
Thank you for your diligent work in improving the manuscript.
Minor spelling issues
Reviewer 3 Report
The authors have diligently and effectively addressed the queries that were raised, resulting in a noticeable enhancement in the overall quality of the manuscript. As a result, I am pleased to recommend that the manuscript be accepted in its current form.